# Numerical Study on the Yaw Control for Two Wind Turbines under Different Spacings

Zhiqiang Xin [1,*], Songyang Liu [1], Zhiming Cai [1], Shenghai Liao [1] and Guoqing Huang [2]

1   College of Mechanics and Materials, Hohai University, Nanjing 211100, China; 201308010015@hhu.edu.cn (S.L.); 211308010001@hhu.edu.cn (Z.C.); 191308040007@hhu.edu.cn (S.L.)
2   School of Civil Engineering, Chongqing University, Chongqing 400044, China; ghuang1001@cqu.edu.cn
*   Correspondence: xinzhiqiang@hhu.edu.cn

**Abstract:** In this study, the large eddy simulation method and the actuator line model are used to investigate the wake redirection of two turbines. Different turbine spacings and yaw-based control of the upstream turbine are considered. The effects of yaw angle and turbine spacing on the output power of two turbines are comprehensively analyzed, and the physical mechanisms of the wake deficit, deflection and interaction are revealed from the distributions of the wake velocity, turbulence intensity and the structures of wake vortices. The results show that the overall power of two turbines is related to the yaw angle of the upstream turbine and the spacing between two turbines. We find yaw angle is the dominant factor in the total power improvement compared to turbine spacing. Still, a large yaw angle causes significant power fluctuations of the downstream turbine. The deficit of wake velocity and the change of output power are determined by the characteristics of the wake flow field, which the yaw control regulates.

**Keywords:** wind turbine wake; large eddy simulation; actuator line model; spacing effect; yaw control





## 1. Introduction

In the practical application of wind turbines, the power loss brought on by the interaction between upstream and downstream turbines is significant for a big wind farm made up of dozens of turbines tightly grouped in an array. When the wind direction is aligned with the arrangement of turbines, the power loss is up to 30–40% [1,2]. Therefore, studying the mutual interference of turbine wake is of great significance. The wake effects within a wind farm have been extensively studied. At present, there are four primary methods for investigating the wake effect of wind turbines: in-site measurements, wind tunnel experiments, wake models and computational fluid dynamics (CFD). Due to the high cost of in-site measurements and wind tunnel experiments, most research on wake mainly relies on the latter two methods.

For the wake model, Katic [3] proposed a one-dimensional linear wake model (Park Wake Model) to analyze the wake effect in wind farms and applied it to WAsP (a wind energy resource assessment software). Based on the one-dimensional wake model, different types of two- and three-dimensional wake models have been put forward. For example, the Ainslie [4] wake model and Larsen [5] wake model are two-dimensional wake models. The three-dimensional wake models based on CFD theory, such as the RGU [6] model, have been proposed by some researchers and are used in wind farm simulation. There are also numerous wake models that take wind plant control into account. For example, Martínez-Tossas et al. [7] proposed the curled wake model to generate a curled wake profile in the wind farm for the yawed wind turbine. However, most wake models are too simplified to capture the turbulence characteristics of wind farm wake, owing to axisymmetric assumptions of the wake and the neglect of wind shear and meandering.

With the rapid development of computer technology, CFD simulations have been widely applied in wind turbine fields, effectively dealing with the three-dimensional

unsteady rotating flow. Newman [8] firstly applied the actuator disk model (ADM) to the evaluation of the turbine wake, and then the large eddy simulation (LES) was combined with ADM to simulate the wake of the wind turbine [9,10]. To improve the accuracy of ADM, Sørensen and Shen et al. [11] proposed the actuator line model (ALM), which can consider the impact of blade geometry. Thus, ALM can be used to investigate the blade tip and root vortices. To evaluate the influence of complex inflow on the operating state of the wind turbine in actual work conditions, Churchfield et al. [12,13] developed Simulator for Wind Farm Applications (SOWFA) based on Open Field Operation And Manipulation (OpenFOAM), which was capable of studying the aerodynamic and wake characteristics of wind turbines in wind farms under the atmospheric boundary layer (ABL) with ALM. In SOWFA [14], the LES method was used to simulate the ABL turbulence and analyze the wake effect of the turbine. The models of the actuator class mainly consist of the actuator disk, the actuator line and actuator surface for the high-fidelity simulation.

Fletcher and Brown [15] studied the wake effect of two turbines with different longitudinal and transverse spacings using the vorticity transport model. The downstream turbine was located entirely or partially in the wake region of the upstream turbine in their study. Mikkelsen et al. [16] combined the actuator line and CFD to study the aerodynamic output power of the wind farm and the mutual effects of the wake of three turbines in the cases that the downstream turbine is fully or partially located in the upstream turbine's wake area. These studies demonstrate that when the spacing between two wind turbines is greater than five times the rotor diameter (D), the interaction between the turbines is reduced and the overall power of the wind farm increases significantly. Similarly, Troldborg et al. [17] analyzed the wake effects of two turbines with different spacings at various inflow conditions using ALM and LES. It was found that when the downstream turbine operates in the wake of the upstream turbine, the average value of blade load decreases, but the standard deviation increases. The aforementioned studies analyze the impact of turbine spacing on downstream turbines and provide a clear explanation of how multi-turbine wake interference works. However, they only consider the influences of wind turbine layout without using control strategies for turbines to improve the performance of the wind farm.

Many scholars have studied the mechanism of yaw control of wind turbines. Munters et al. [18] studied six kinds of wind farm layouts and analyzed the effects of induction control, yaw control and combined induction-yaw control. It was found that yaw control is suitable for the aligned layout wind farm. Zong et al. [19] systematically studied the physical characteristics and effectiveness of active yaw control under different wind conditions with wind tunnel experiments and a newly developed analytical model of wind farms. They found that the spacing of turbines and other factors would influence active yaw control. When there are multiple turbines, the optimal yaw angle continuously decreases from upstream to downstream turbines, which is related to the secondary wake steering effect. Ma et al. [20] discovered that by using a cooperative yaw control strategy, the total output power of five aligned turbines can be increased by 17.5 percent. Paul Fleming et al. [21] used high-fidelity simulations to investigate the effects of the yaw and tilt angles of an upstream turbine on the downstream turbine without changing the position of the upstream turbine. Bastankhah et al. [22] suggested that when the spacing between two turbines is 5D, the total power output of two turbines will not continuously increase with the increasing of yaw angle of the upstream turbine, and the optimal yaw angle is 30°.

Based on the above research, we know that the wake of upstream turbines would strongly effect downstream turbines, resulting in the deflected velocity deficit, counter-rotating vortices, etc. Hence, it is needed to adopt appropriate yaw control strategies to improve the performance of wind farms. However, most previous research does not comprehensively consider the joint action of the spacing between two turbines and the yaw angle of the upstream turbine. It is conceivable that when the yaw control is adopted for the upstream turbine, the wake of the upstream turbine will be correspondingly deflected.



It is necessary to investigate whether yaw control strategies based on fixed spacing are still effective or obvious when the spacing between two turbines varies. In this paper, the combined influences of yaw angle and turbine spacing will be fully analyzed by comparing the output powers and wake effects of two turbines in different yaw angles and longitudinal spacings.

## 2. Physical Models and Numerical Approach

### 2.1. Large-Eddy Simulation

In this study, the LES method is used to simulate the wake turbulence of the wind turbines. The incompressible fluid is divided into large-scale and small-scale eddies by filtering, which are calculated, respectively. The resolved-scale (large-eddy scale) dynamics are obtained by solving the spatially filtered, incompressible Navier–Stokes equations, while the small scales are resolved implicitly by the sub-grid scale (SGS) model. It consumes fewer computing resources than direct numerical simulation (DNS), but can achieve higher accuracy than Reynolds-Averaged Navier Stokes (RANS) simulations [23].

The filtered momentum equation is [24]:

$$\begin{cases} \frac{\partial \bar{u}_i}{\partial t} + \frac{\partial (\bar{u}_i \bar{u}_j)}{\partial x_j} = -\frac{1}{\rho}\frac{\partial \bar{p}}{\partial x_i} + \nu \frac{\partial^2 \bar{u}_i}{\partial x_i \partial x_j} - \frac{1}{\rho}\frac{\tau_{ij}}{\partial x_j} + \frac{1}{\rho} f_i \\ \frac{\partial \bar{u}_i}{\partial x_i} = 0 \end{cases} \tag{1}$$

where $u$ is the filtered velocity, $\bar{p}$ is the filtered pressure, $t$ is the time, $\rho$ is the air density which is about 1.225 kg/m$^3$, $\nu$ is kinematic viscosity, $\tau_{ij}$ refers to the sub-grid scale (SGS) stress, $f_i$ is the force exerted by the actuator line turbine model.

The simulations in the present study are carried out with SOWFA, which is developed by Churchfield et al. [12,13] based on OpenFOAM v-2.4.x. The pressure-implicit with splitting of operators (PISO) method is used for solving the momentum equation. Time and space discretization are performed in the Backward format and Gaussian linear format.

### 2.2. Actuator Line Model

ALM is a full three-dimensional transient aerodynamic model first proposed by Sørensen and Shen [11]. In order to reproduce the effect of the turbines in a wind farm, the wind turbines are represented as body force fields. In ALM, the rotating blades are simplified into actuator lines, which several blade elements are distributed along. Once the aerodynamic forces at each blade element are calculated, the corresponding forces are projected onto the flow field to exert the effect of the turbine blades. For the local wind speed, the aerodynamic volume force [25] is solved by:

$$f_e = \frac{1}{2}\rho U_{rel}^2 c(C_L e_L + C_D e_D)^T \tag{2}$$

where $U_{rel}$ is the local wind velocity, $c$ is the chord length of the airfoil, $C_L$ and $C_D$ are the lift and drag coefficients, and $e_L$, $e_D$ are unit vectors in the direction of lift and drag force, respectively.

The tip/root loss correction was implemented by the Prandtl–Glauert rule, and the specific formula of correction factor $F$ [26] was as follows:

$$F = \frac{4}{\pi^2} \cdot \arccos\left[\exp\left(-\frac{B}{2}\frac{R_{tip} - r}{r\sin\phi}\right)\right] \cdot \arccos\left[\exp\left(-\frac{B}{2}\frac{r - R_{root}}{r\sin\phi}\right)\right] \tag{3}$$

where $B$ is the number of the blades; $R_{tip}$ and $R_{root}$ are the radius of the blade tip and root; $r$ is the spacing between the blade-element location and the root of the blade; and $\phi$ is the angle between the local relative velocity and the chord.

The actuator forces are projected to body forces through using a three-dimensional Gaussian function:

$$f(a) = \frac{1}{\varepsilon^3 \pi^{\frac{3}{2}}} \exp\left[-\left(\frac{a}{\varepsilon}\right)^2\right] \tag{4}$$

where $a$ is the spacing between the body force center and the projection point, and $\varepsilon$ is the smoothing factor, which controls the numerical stability and physical sense of the projection method. Troldborg et al. [17] pointed out that it is appropriate when $\varepsilon$ is twice the grid length.

### 2.3. Control Strategy of Wind Turbine

Due to the influence of the yaw and wake velocity deficit, the turbines cannot maintain the rated working state, so it is necessary to adopt autonomous control to adjust the angular speed of the turbines. The speed control of the rotor is carried out by using the Five-Region Method [27], as shown in Figure 1, in which the relationship between the angular rotor speed and the generator torque is specified.

$$\frac{d\Omega}{dt} = \frac{1}{I_{Drivertrain}}(T_{Aero} - N_{Gear}T_{Gen}) \tag{5}$$

where $\Omega$ is the angular velocity of the rotor, $T_{Aero}$ is the torque of the turbine, $T_{Gen}$ is the torque of the generator, which is obtained through the control of the Five-Region method, $N_{Gear}$ is the speed ratio of the gearbox, $I_{Drivertrain}$ is the total rotational inertia of the transmission system.

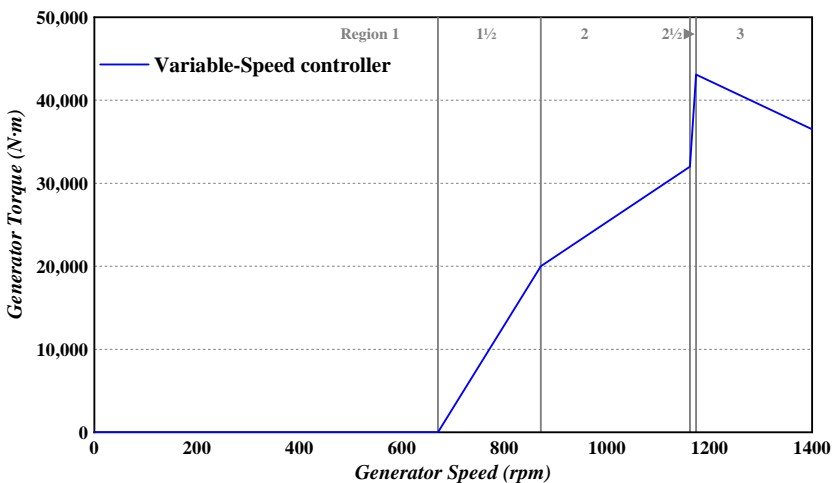

**Figure 1.** Torque-versus-speed response of the variable-speed controller.

When the wind speed is greater than the rated wind speed, the wind turbine turns on the pitch control based on the PID algorithm. In this study, the change of pitch angle during the control process is as follows:

$$\Delta\theta = K_P N_{Gear}\Delta\Omega + K_I \int_0^t N_{Gear}\Delta\Omega dt \tag{6}$$

where $K_P$ is the proportional gain, $K_I$ is the integral gain, $\Delta\Omega$ is the speed deviation value, and $N_{Gear}$ is the gear speed ratio.

### 2.4. Validation of the Numerical Methods

The wind turbine that was used in this study is the National Renewable Energy Laboratory's (NREL) 5 MW reference turbine. The standard parameters of NREL-5MW are clearly introduced by Jonkman et al. [27], and some characteristic parameters are shown in Table 1.

**Table 1.** NREL-5MW standard turbine parameters.

|  | NREL-5MW |
| --- | --- |
| Rating | 5 MW |
| Rotor Orientation | Upwind |
| Number of blades | 3 |
| Rotor Diameter | 126 m |
| Hub Height | 90 m |
| Rating Wind Speed | 11.4 m/s |
| Rating Rotor Speed | 12.1 RPM |

In order to verify the accuracy of the numerical methods, the wake of two turbines with a spacing of 7D under the neutral ABL inflow is simulated, according to Churchfield's research [13]. A precursor simulation is carried out to set up the neutral wind field. The surface aerodynamic roughness height, $z_0$ is set at 0.001 m, which is a typical offshore condition. The horizontally averaged wind speed at the hub height of 90 m is driven to 8 m/s.

In the precursor inflow simulation, the total physical time is 21,000 s, and the flow fields in the last 3000 s are extracted as the inflow boundaries for the wake simulation. The neutral atmospheric wind field, generated by the precursor inflow simulation, is shown in Figure 2a,b represents the velocity field of the vertical plane in the center line of the wind field. It can be seen from Figure 2 that, due to the influence of ground roughness, the wind speed close to the surface is low, and the wind speed increases in the height direction.

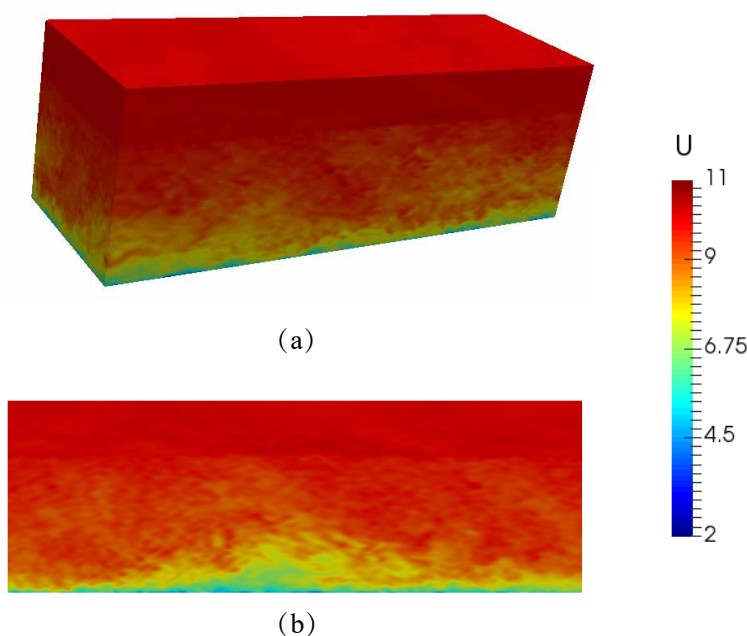

(a)

(b)

**Figure 2.** The wind field in the precursor simulation: (**a**) overview view; (**b**) vertical plane.

For wake simulation, the NREL-5MW turbine is used for simulation in this section. The grid size around the turbine is set to be 2.5 m and the calculation time step is set to be 0.025 s. The velocity deficit profiles in the hub height plane and longitudinal plane are shown in Figure 3. The first profile shown is 1D upstream of the first turbine, and each subsequent profile is 1D downstream of the previous one. The velocity deficit scale for each profile ranges from 0 to 0.5. The time-averaged wake velocity deficit profiles are normalized by the hub-height average freestream speed. From Figure 3, the present results are in good agreement with Churchfield's research [13]. This shows that the numerical methods of the wind turbine wake are accurate and effective.

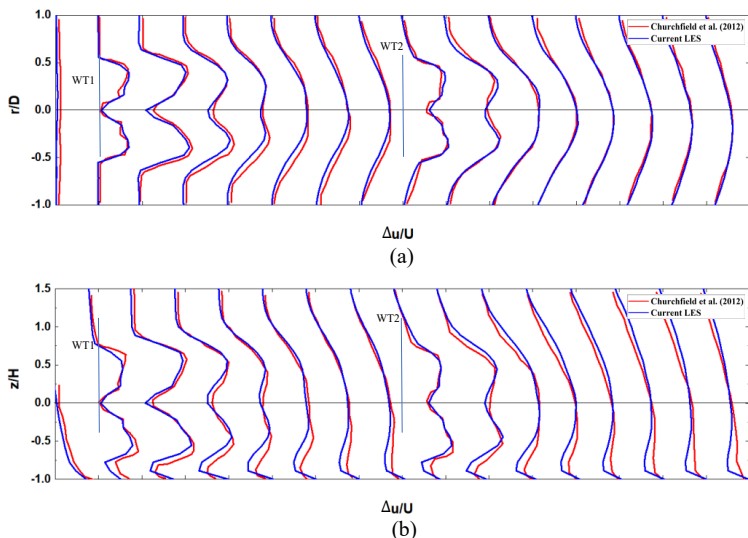

**Figure 3.** Time−averaged wake velocity deficit profiles normalized by the hub-height average freestream speed of 8 m/s: (**a**) horizontal profiles at hub height; (**b**) vertical profiles in the wake centerline [13].

### 3. Simulation Setting

In this study, all simulations are investigated with two turbines, which are arranged on the central axis of the wind field as shown in Figure 4. The simulation domain is 2000 m long, 500 m wide and 500 m high. In order to save computing costs, the SnappyHexMesh refinement method of OpenFOAM is carried out on the grid. The uniform hexahedron grids with a side length of 2.5 m are used in the central computing area, and the outer grid sizes are 5 m and 10 m, respectively. The total number of grids is about 13 million. As Li et al. [28] selected uniform inflow in his research, an idealized uniform inflow is used in our study to focus only on the effects of yaw control and spacing, and the wake fields under the turbulent inflow with the turbine spacing of 7D are investigated to analyze the influences of the turbulent intensity on the yaw control. The uniform inflow speed is 11.4 m/s. Similar to Munters' [29] studies, the symmetry boundary conditions are specified on the top, bottom, left and right sides. The hub of the turbine is placed in the middle of the height direction of the calculation domain to minimize any influence of vertical domain boundaries. The turbines are arranged in the midline of the calculation domain, so that the rotor axis is located at the center of the flow field. The upstream turbine is 250 m away from the inlet and the downstream turbine is about 5–8D away from the upstream turbine.

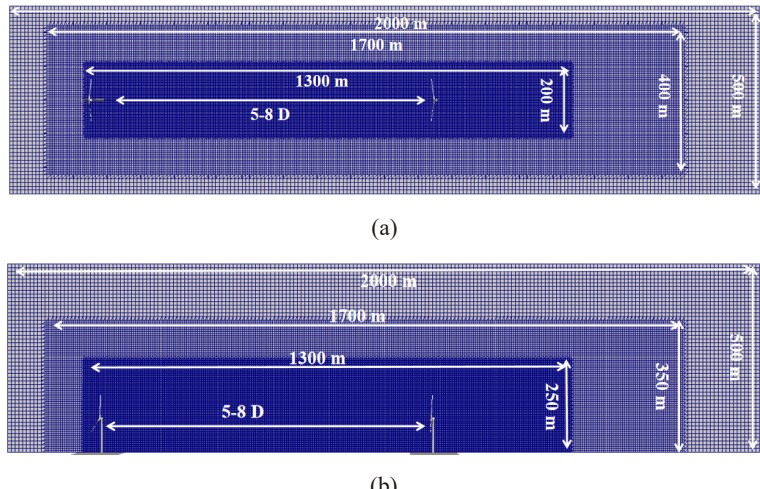

**Figure 4.** Schematic diagrams and grids of the calculation domain: (**a**) Top View; (**b**) Side View.

Churchfield [13] pointed out that it was needed to use at least 20 grid cells across the rotor diameter. In this paper, we place 50 grid cells across the rotor to ensure an accurate simulation of the tip and root vortices. The time step is given according to the Courant–Friedrichs–Lewy (CFL) criterion, that the Courant number (*Co*) must be less than 1.

$$Co = \frac{u\Delta t}{\Delta x} \tag{7}$$

where $u$ is the local wind speed, which is the vector sum of inflow velocity and blade tip velocity. $\Delta x$ is the grid size, and $\Delta t$ is the time step. According to the Courant number criterion, $\Delta t$ needs to be less than 0.031 s. A series of cases with different mesh resolutions and time are investigated to verify the accuracy of the simulation, as shown in Table 2. We choose the output power of the first turbine and the wake velocity together as the judgment bases for the independence verification. For the five cases, the velocity profiles at hub height at 2D downstream of the downstream turbine are shown in Figure 5. After a comprehensive consideration of computation cost and precision, the mesh length in the dense area is finally chosen as 2.5 m, and the time step is taken as 0.025 s.

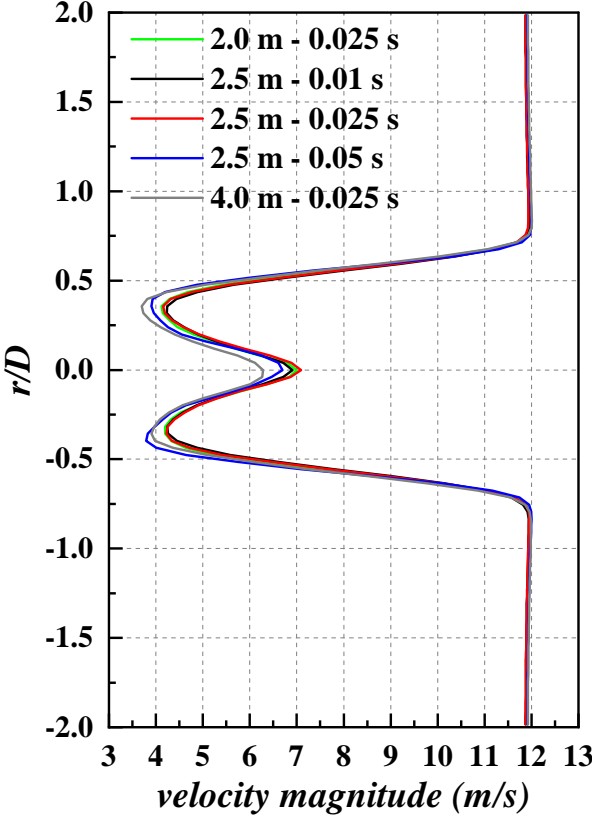

**Figure 5.** Velocity profiles of the hub height plane at 2D downstream of the downstream turbine.

**Table 2.** Grid and time step independent validation.

| Situation | Mesh Length (m) | Time Step (s) | Power (MW) |
|:---:|:---:|:---:|:---:|
| 1 | 2.0 | 0.025 | 5.302 |
| 2 | 2.5 | 0.01 | 5.299 |
| 3 | 2.5 | 0.025 | 5.298 |
| 4 | 2.5 | 0.05 | 5.185 |
| 5 | 4.0 | 0.025 | 5.113 |

## 4. Results

### 4.1. Angular Speed of the Rotors

From the previous studies about the wake redirection control [26,30], the yaw angle is generally in the range of 0° to 30°. Therefore, when we investigate the effect of the yaw condition, the yaw angles are taken as 0°, 10°, 20° and 30°, respectively.

The rotor speeds of two turbines are adjusted by the Five-Region Method and tend to a dynamic balance, respectively. Figure 6 shows the average rotor speed of each turbine in different cases. For a specific turbine spacing, with the increase in yaw angle, the rotor speed of WT1 (the upstream wind turbine) decreases while that of WT2 (the downstream wind turbine) increases. Under a specific yaw angle, the turbine spacing mainly affects the operation state of WT2. With the increase of turbine spacing, the rotor speed of WT2 also increases, but the growth rate of this speed with turbine spacing is not linear. In comparison, the effect of yaw angle on rotor speed is greater than that of turbine spacing and at the 20° yaw angle, the effect of spacing on WT2 is more obvious than at the other yaw angles.

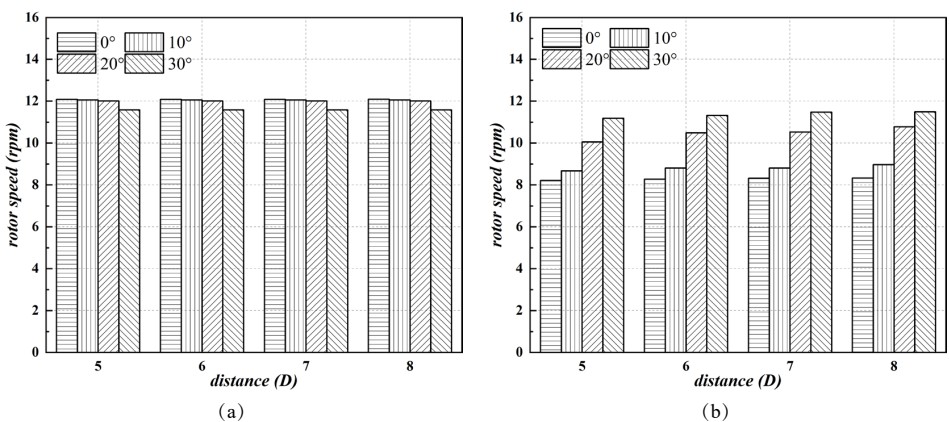

**Figure 6.** The average rotor speed of each turbine in different cases: (**a**) WT1; (**b**) WT2.

### 4.2. Instantaneous Power Analysis of Two Turbines

The power-time curves of both turbines are shown in Figure 7. It is evident from it that, shortly after the simulation begins, WT1 and WT2 operate at similar power, but when the upstream wake reaches WT2, the output power of WT2 begins to decrease. After a period of time, both turbines run steadily at the corresponding speed and maintain a relatively stable output power, and the power curves eventually present a small range of periodic fluctuations. As shown in Figure 7a, without yaw control, the output power of WT2 for 8D is not significantly higher than that for 5D, 6D and 7D, which indicates that the wake velocity still does not recover apparently even at 8D downstream.

When the yaw angle of WT1 is zero, the output power of WT1 is the biggest. However, WT2 is completely located in the wake region, which makes the output power of WT2 reach its minimum number and the average power of WT2 increase with the increase of yaw angle of WT1, but the power fluctuation of WT2 becomes larger.

In this study, if the inflow condition is not changed, the operating state of WT1 is only related to its own yaw angle. The yaw control makes the shaft of the turbine deviate from the direction of wind, reducing the windward area and thereby leading to the decrease in the output power of WT1. As the yaw angle of WT1 enlarges from 0° yaw to 10°, 20° and 30°, the output power of WT1 decreases by 3.36%, 11.83% and 25.22%, respectively, which is similar to the power loss of the first turbine in Churchfield's research [30], where he studied the yaw control with five turbines. However, WT1 performs similarly at various spacings with the same yaw angle. WT2 can only affect the far wake area of WT1 rather than its near wake area. The output power of turbines is mainly determined by the local wind field near the rotors, so the output power of WT1 is not affected by WT2.

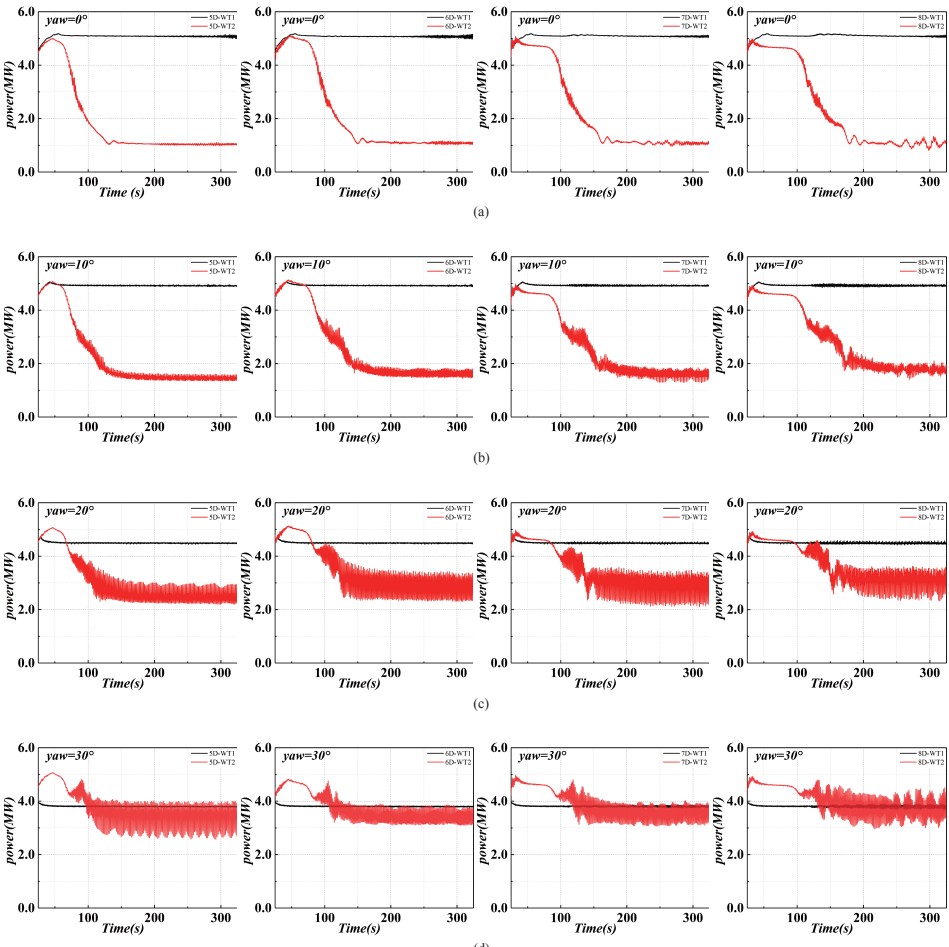

**Figure 7.** Power-time curves of two turbines with different spacings and upstream yaw angles: (**a**) yaw = 0°; (**b**) yaw = 10°; (**c**) yaw = 20°; (**d**) yaw = 30°.

The upstream wake is deflected by adopting yaw control, and the output power of WT2 is significantly improved. In addition, due to the asymmetric wake from WT1, the power fluctuation range of WT2 observably increases, and the fluctuation range positively correlates with yaw angle. This may lead to the reduction in the quality of output electricity and also increase fatigue loading of the turbine and shorten its working life.

The dimensionless standard deviation of the output power of WT2 is evaluated to analyze the power fluctuation, and it is given by:

$$SD = \frac{\sqrt{\left(P_i - \bar{P}\right)^2}}{\bar{P}} \tag{8}$$

where $P_i$ is the instantaneous output power of WT2, and $\bar{P}$ is the average value of the output power of WT2 in the stable period.

The standard deviations of the power fluctuation of WT2 under various working conditions are shown in Figure 8. The power fluctuation of WT2 is related to the asymmetry of the wake caused by WT1 and the wake expansion. The wake asymmetry depends on the yaw angle and turbine spacing, but the wake expansion only depends on the turbine spacing. It can be found that when the spacing between two turbines is small, such as in the case of 5D, the output power fluctuation of WT2 increases with the yaw angle. When the yaw angle of WT1 is 0°, 10°, 20° and 30°, the standard deviation of the output power fluctuation of WT2 for 5D is 3.64%, 6.17%, 9.59% and 11.07%, respectively. When the spacing is 6D, 7D and 8D, the fluctuation for the yaw angle of 30° is smaller than that for the yaw angle of 20°.

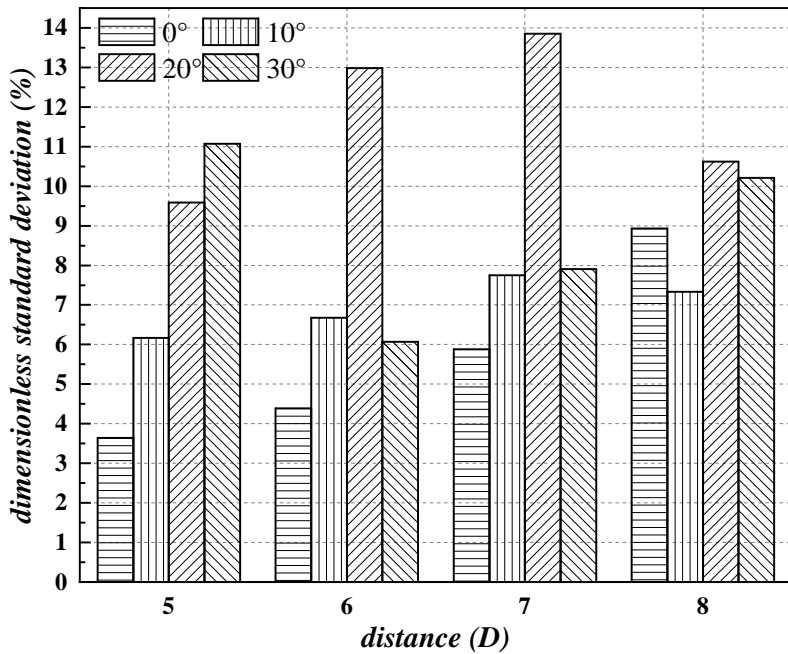

**Figure 8.** Standard deviations of the power fluctuation of WT2 under various working conditions.

When the yaw angle is 0°, the standard deviation of the output power fluctuation of WT2 for the cases of 5D, 6D, 7D and 8D is 3.64%, 4.39%, 5.88% and 8.93%, respectively, due to the wake expansion. When the yaw angle of WT1 becomes larger, in the dual role of the asymmetric distribution and spread of wake, the standard deviation of the output power of WT2 no longer increases monotonously. From the above analysis, it can be concluded that the influence of yaw angle on power fluctuation is more significant than that of turbine spacing.

### 4.3. Average Output Power Analysis

In order to evaluate the total output power of two turbines clearly, the average output power of the turbines will be analyzed in this section as shown in Figure 9.

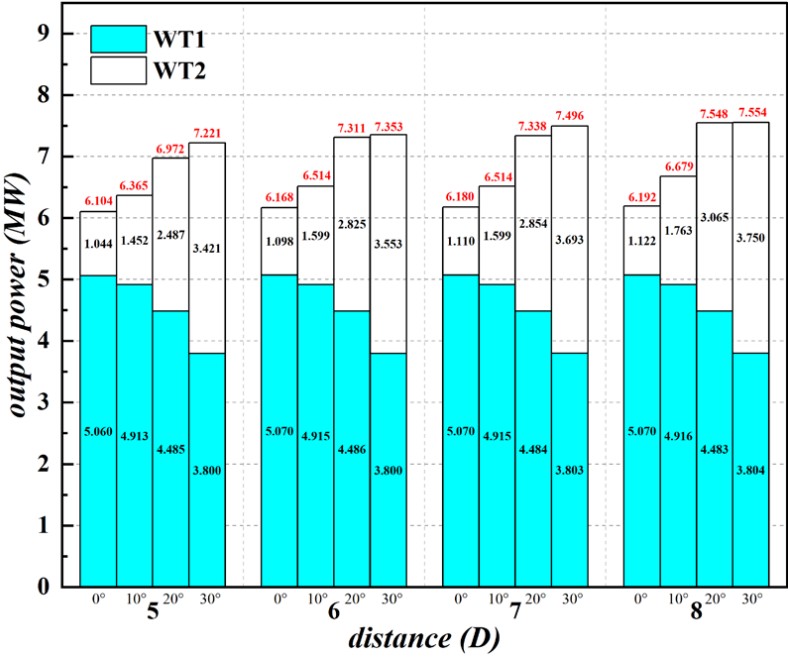

**Figure 9.** Average output power of the turbines.

When the yaw angle increases from 0° to 30°, the output power of WT1 gradually decreases while the output power of WT2 increases. In the range of 0° to 20°, the total power of two turbines shows an upward trend. When the yaw angle increases from 20° to 30°, the total power of two turbines with the spacing of 5D, 6D and 7D still increases, but it does not change significantly when the spacing is 8D.

The relationship between total power and the spacing of two turbines is also discussed here in different yaw angles. It can be found that when the yaw angle is 0°, the total power does not increase significantly when the spacing between two turbines increases from 5D to 8D. At the yaw angle of 0°, WT2 has a large loss of output power as a result of being fully affected by the wake of the upstream turbine.

When the yaw angles are 10° and 20°, there is a significant increase in the total power from 5D to 6D. Then, the improvement of the total power is not obvious from 6D to 7D, but the total power increases again from 7D to 8D. When the yaw angle is 30°, the total power increases slightly with the increase in spacing. Under the fixed spacing, the yaw control of WT1 has a significant impact on the increase in the total power. For the 5D, 6D, 7D and 8D cases, the total power with a 30° yaw angle increases by 18.31%, 19.20%, 21.29% and 21.99%, respectively, compared to a 0° yaw angle. However, at the same time, it can be found that the increase in yaw angle does not have a simple monotonic relationship with the increase in total power. When the spacing is 8D, the total power under the yaw angle of 20° and 30° is similar, this phenomenon can also be found in the study of Dong [31]. For the 0°, 10°, 20° and 30° cases, the total power with a 8D spacing increases by 1.45%, 4.93%, 8.26% and 4.61%, respectively, compared to a 5D spacing, the increase in spacing has the most obvious effect on the increase in total power under 20° cases. In general, the total power of two turbines is related to the yaw control of WT1 and the spacing. When the spacing is within 8D, the yaw control of WT1 plays a more significant role in improving the total power.

*4.4. Wake Time-Averaged Velocity*

For all points in the calculation domain, the quasi-periodic stable velocities in the last 200 s of wake simulations are averaged.

Figure 10 shows the time-averaged wake velocity deficit profiles normalized by the hub-height average speed. It can be found from Figure 10 that the characteristics of the wake velocity of WT1 are only related to the yaw angle. As shown in Figure 10a, when the yaw angle is 0, the wake velocity profiles on the left and right sides of the turbine present a high degree of symmetry. The average velocity profiles become more chaotic as the yaw angle increases, as shown in Figure 10b–d. However, a comprehensive comparison of Figure 10 shows that with the increase in yaw angle, the velocity deficit behind WT2 decreases. It can be seen that the increase in yaw angle can significantly increase the inflow velocity of WT2, and the position of bimodal shape of WT1 gradually shifts away from the centerline where $y/D = 0$.

Clearly, the wake velocity profiles under different turbine spacing are different only after WT2, and the difference becomes more evident with the increase in yaw angle. As shown in Figure 10a, for the 0° yaw angle, there is little difference in the velocity deficit profiles at 10D downstream of WT1.

Generally speaking, the recovery of the wake velocity deficit is closely related to the yaw angle. Yaw control will lead to the offset of the wake centerline. The greater the yaw angle is, the greater the offset of the wake centerline in front of WT2 will be. This situation will change after passing through the downstream turbine. In addition, the larger the spacing between two turbines is, the greater the inflow wind speed of WT2 will be, because the impact of WT1's wake on WT2 is relatively weakened.

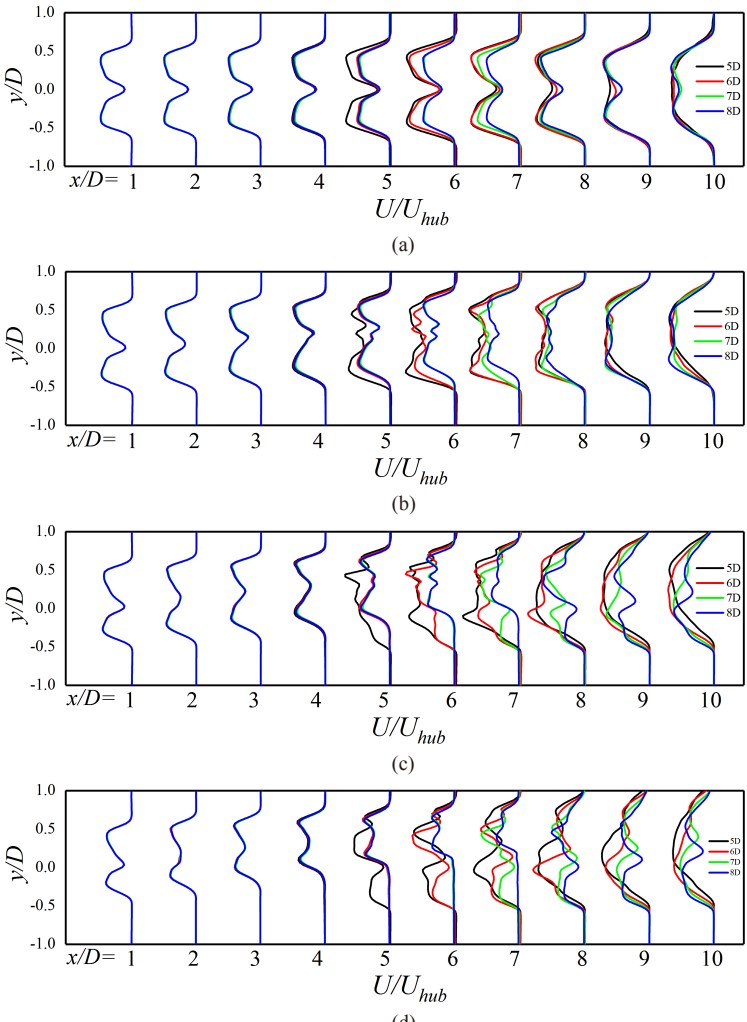

**Figure 10.** Time−averaged wake velocity deficit profiles normalized by the hub−height average speed of 11.4 m/s. The velocity deficit scale for each profile ranges from 0 to 1. Each consecutive profile is taken 1D downwind: (**a**) yaw = 0°; (**b**) yaw = 10°; (**c**) yaw = 20°; (**d**) yaw = 30°.

The contours of time-averaged velocity are shown in Figure 11 in a horizontal plane at hub height. The change in spacing from 5–8D has no obvious effect on the velocity deficit of WT1 wake under uniform inflow. However, the yaw control can obviously change the wake direction of WT1. Wake redirection makes the wake of WT2 present an asymmetrical distribution, and the wake velocity deficit decreases. Under the conditions with large yaw and spacing, the range of low velocity region in the downstream wake has shrunk obviously.

In addition, it can be found that when the yaw angle of WT1 is as large as 30°, the maximum wake velocity deficit will not appear immediately on the left side (opposite to the wake deflection induced by the yaw control) behind WT2 but after some distance, and then it slowly recovers.

From Figure 11, we can find that when the yaw angle reaches 20°, WT2 is still under strong wake effects because the spacing between two turbines is relatively small under 5D, 6D and 7D conditions. Moreover, the power rise of WT2 is greater than the power loss of WT1 when yaw changes to 30°, so total power is still rising. For 8D situations, the influence of WT1 on the wake field of WT2 has been weakened when the yaw angle is 20°, so the change of the total output power is not evident under yaw control from 20° to 30°.

In the working condition with the yaw angle of 30° and spacing of 7D, the overall performance of two wind turbines is relatively optimal. The wake velocity contours of two

turbines in yz slices at different locations downstream of two wind turbines corresponding to this case, as shown in Figure 12, are chosen to be analyzed. Figure 12a shows the velocity contour in yz slices at 1D downstream of WT1. It can be found that the wake shape has changed from a circular shape to an elliptical shape due to the wake deflection. Figure 12b,c show the velocity contours at 4D and 6D downstream of WT1. It can be found that the wake becomes curled during the wake evolution, which illustrates the curled nature of the wake with yaw control. Figure 12d–f represent the velocity contours at 1D, 2D, and 4D downstream of WT2. Which show that the strength of the overlap wake in the downstream of WT2 is asymmetric, and the wake velocity gradually recovers with the increase in distance.

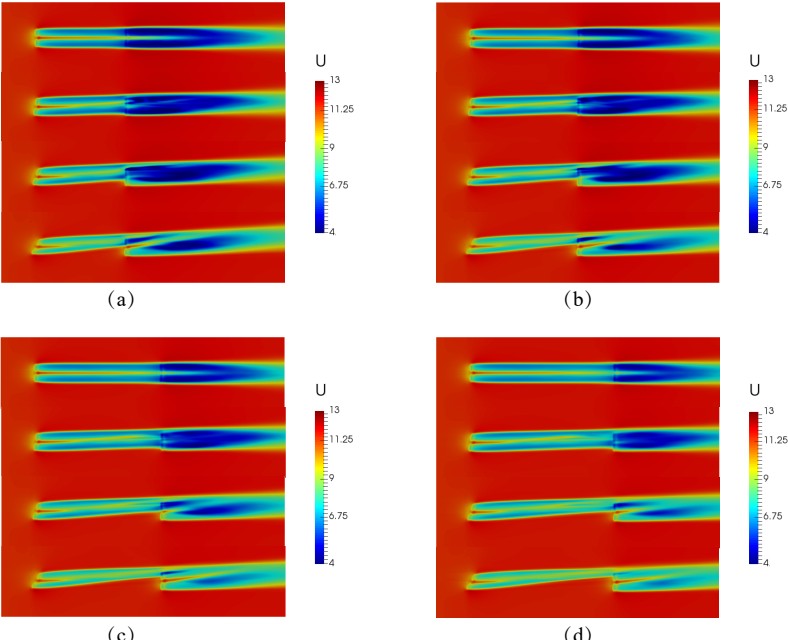

**Figure 11.** Contours of time-averaged velocity in the horizontal plane at hub height: (**a**) spacing = 5D; (**b**) spacing = 6D; (**c**) spacing = 7D; (**d**) spacing = 8D.

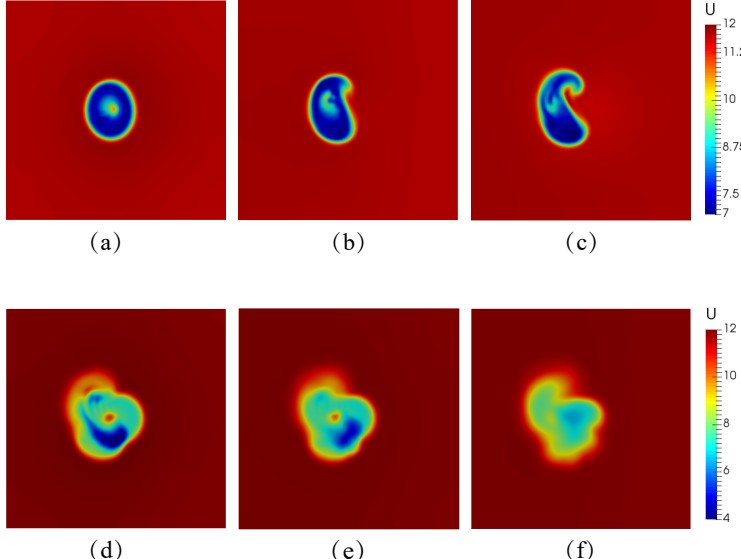

**Figure 12.** Contours of time-averaged velocity in the vertical plane at hub height: (**a**) 1D downstream of WT1; (**b**) 4D downstream of WT1; (**c**) 6D downstream of WT1; (**d**) 1D downstream of WT2; (**e**) 2D downstream of WT2; (**f**) 4D downstream of WT2.

### 4.5. Turbulence Intensity

Figure 13 shows the profiles of wake turbulence intensity from 1–10 D downstream of WT1. The turbulence intensity ($I$) is calculated by:

$$I = \sqrt{\frac{1}{3U_{hub}^2}\left(\langle u'^2 \rangle + \langle v'^2 \rangle + \langle w'^2 \rangle\right)} \tag{9}$$

where $u'$, $v'$ and $w'$ are the fluctuating velocity components in the x, y and z axis, respectively. $U_{hub}$ is the speed at the hub height.

The turbulence intensity of far field wake is related to the spacing between the turbines; the smaller the spacing is, the greater the intensity will be. The increase in turbulence intensity is only about 10% behind WT1, while the turbulence intensity increases to 25% after passing through WT2. When the spacing is small, the turbulence intensity can even reach 30%. The yaw control also has a great influence on the turbulence intensity of the far wake.

Figure 14 is the turbulence intensity contours. It can be seen from Figure 14 that the spacing of turbines has an influence on the turbulence intensity. The closer two turbines are, the greater the turbulence intensity of WT2's wake and the wider the turbulence influence range. When the yaw angle of WT1 is 0°, the turbulence intensity and turbulence diffusion range of WT2 wake are the largest. With the increase in the yaw angle of WT1, the turbulence intensity after WT2 wake can be greatly reduced.

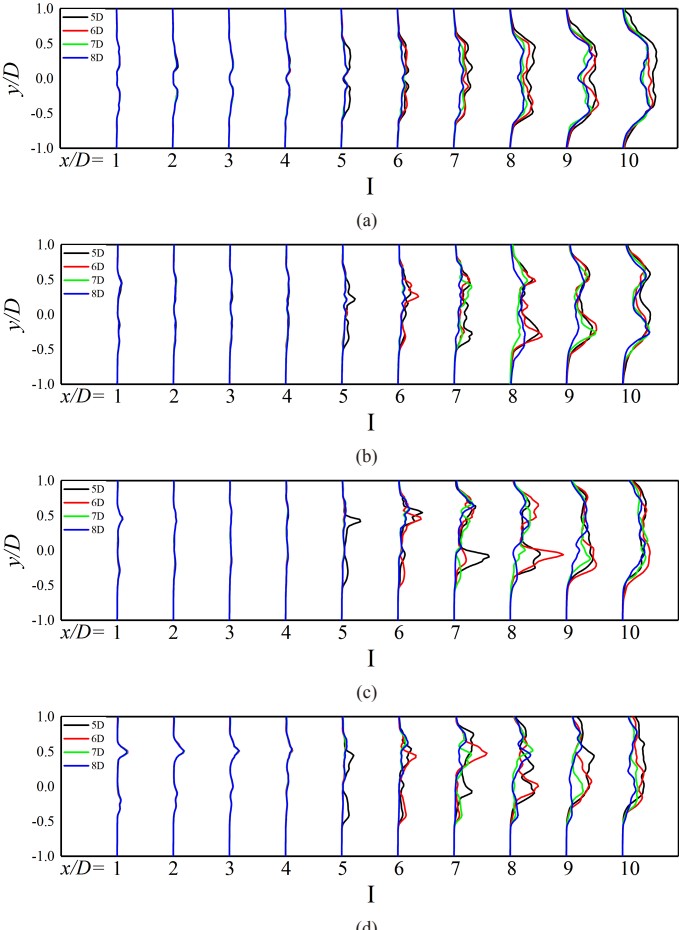

**Figure 13.** Time−averaged turbulence intensity profiles. The turbulence intensity scale for each profile ranges from 0 to 0.5. Each consecutive profile is taken 1D downwind: (**a**) yaw = 0°; (**b**) yaw = 10°; (**c**) yaw = 20°; (**d**) yaw = 30°.

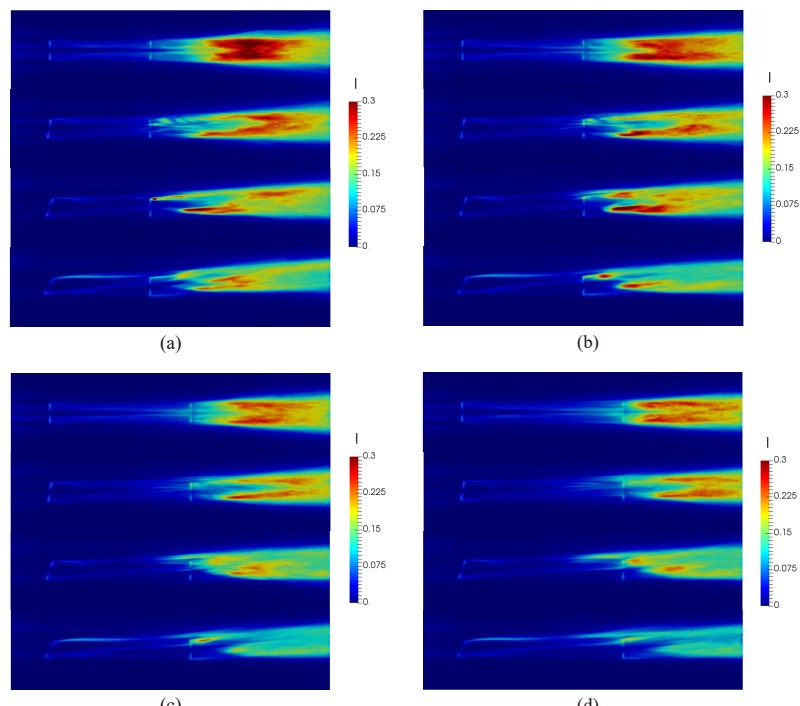

**Figure 14.** Contours of instantaneous turbulence intensity in a horizontal plane at hub height: (**a**) spacing = 5D; (**b**) spacing = 6D. (**c**) spacing = 7D; (**d**) spacing = 8D.

Figure 15 shows turbulence intensity in yz slices at different locations downstream of two wind turbines when the yaw angle is 30° and spacing is 7D, similar to Figure 12. Figure 15a–c depict turbulence intensity contours at positions 1D, 4D and 6D downstream of WT1. It can be seen that the turbulence intensity is smaller than those at the positions downstream of WT2, and the areas with large turbulence intensity are caused by the wake curls and deflection, and the changes of turbulence intensity also present a three-dimensional characteristic. Figure 15d–f are turbulence intensity contours at 1D, 2D and 4D downstream of WT2, respectively. The distributions of turbulence intensity at the distances downstream of WT2 are asymmetric and higher due to the wake superposition of two turbines, and tend to be uniform in the far wake.

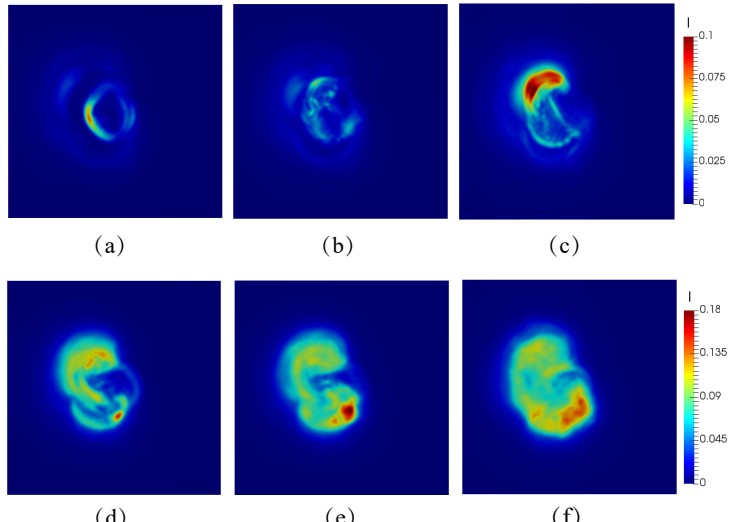

**Figure 15.** Contours of instantaneous turbulence intensity in a vertical plane at hub height: (**a**) 1D downstream of WT1; (**b**) 4D downstream of WT1; (**c**) 6D downstream of WT1; (**d**) 1D downstream of WT2; (**e**) 2D downstream of WT2; (**f**) 4D downstream of WT2.

### 4.6. Vortex Structures

The total power is almost the same when turbine spacing is 7D and 8D, which is consistent with the study of Yu et al. [32]. Considering the cost of power transmission and the site utilization of wind farms, 7D is recommended. The vortices structure of the turbine at 7D is analyzed here in detail.

The evolution of the wake vortices is illustrated through the vortex structures depicted in Figure 16. The wake vortices are identified by the Q criterion. The blade root-bound and tip vortices are clearly visible for WT1, and it moves downward in a spiral shape after shedding from the turbine. With the increase in spacing, the wake vortices gradually diffuse and break up, and they are further disturbed by the blades after reaching WT2, and the complexity of the wake vortices structure increases.

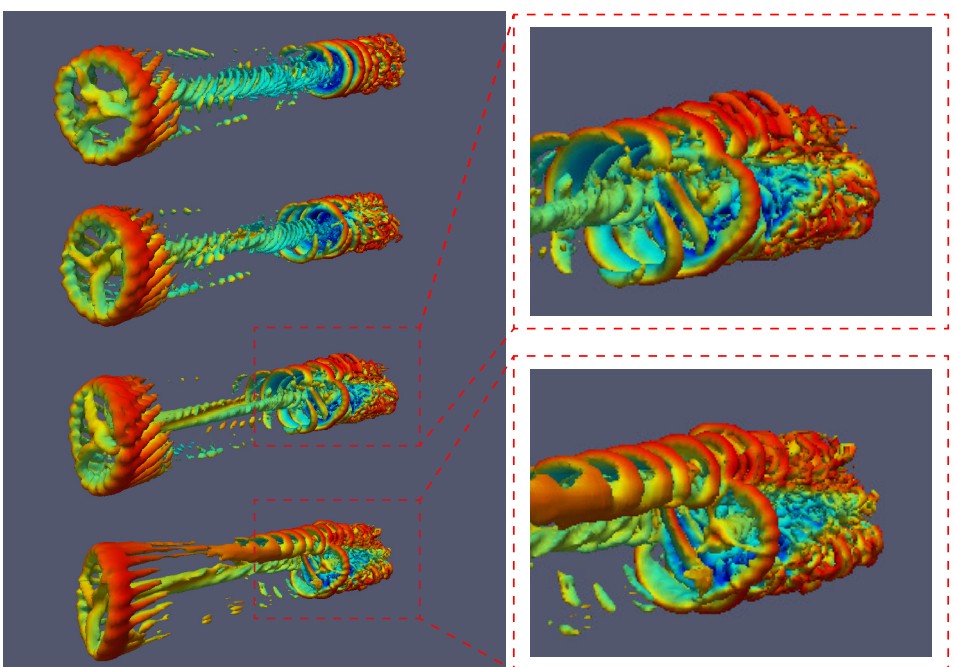

**Figure 16.** Three-dimensional vortices structure in the vicinity of the turbines with 7D spacing for yaw angle from 0° to 30°.

As yaw-based control is adopted, the results show that the vortices shift obviously and have an asymmetric distribution. The vortices shedding from WT1 are crushed when reaching WT2. The tip vortices of WT1, which quickly break down and mix with each other, do not remain well compared with the blade root vortices of WT1. However, it can be observed that there are relatively clear tip vortices in the non-overlapping part of WT2 when the yaw angle of WT1 becomes large. The influence of vibration and uneven inflow caused by asymmetric turbine wake on the operation of downstream turbines should be considered in the actual wind field design.

### 4.7. Turbulent Inflow

In previous sections, we analyzed the overall performance of two turbines for different yaw angles and turbine spacing in uniform inflow. There is also the atmospheric background turbulence effect in a real wind farm [29], which will affect the wake field development, especially in the far field region. Therefore, in this section, the wake fields of two turbines with a spacing of 7D in the ABL wind field are investigated to analyze the effects of the turbulent intensity on the yaw control.

In this section, the precursor simulation of the wind field is carried out, similar to that in Section 2.4. The ground roughness $z_0$ is still set as 0.001. In order to compare with the results of uniform inflow, we set the reference speed at the hub height to 11.4 m/s. With the

inflow boundaries generated by the precursor wind field simulation, the wake fields of two turbines with yaw angles of 0°, 10°, 20° and 30° are simulated. The instantaneous velocity contour in a horizontal plane at hub height is given in Figure 17. It shows that the trend of the velocity deficit corresponding to the neutral ABL inflow is the same as that of uniform inflow. The distribution of velocity in the region far away from the turbine is still uneven, and the wake meandering is quite distinct. In contrast to the uniform inflow, the wind velocity recovers faster after the flow goes over wind turbines. Figure 18 shows the contour of turbulence intensity in the corresponding horizontal plane. The regions with high turbulence intensity are still induced by the wake deflection and interference. The quantity of the turbulence intensity due to the background turbulence is more significant than that of the uniform inflow.

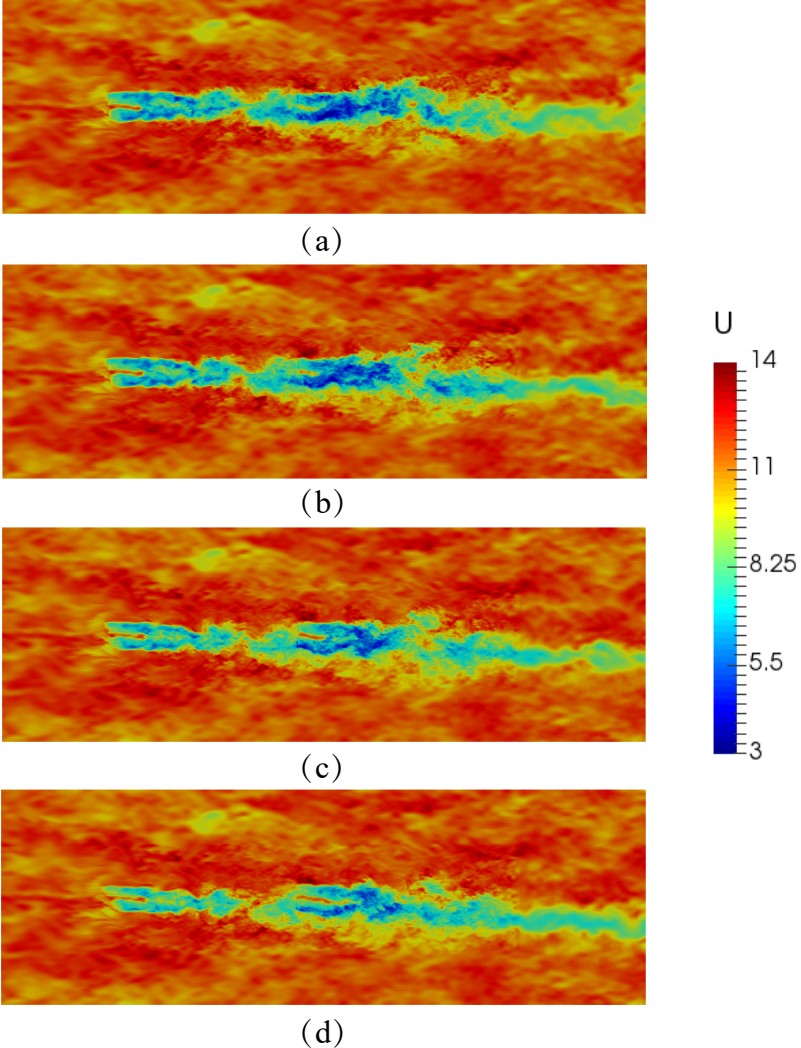

**Figure 17.** Contours of instantaneous velocity in the horizontal plane at hub height: (**a**) yaw = 0°; (**b**) yaw = 10°; (**c**) yaw = 20°; (**d**) yaw = 30°.

The average output power of turbines in the ABL wind field is compared with those of the uniform inflow in Figure 19. It can be found that when there is background turbulence, the power of WT1 is about 5 MW at 0° yaw angle and about 3.8 MW at the yaw angle of 30°. The output power of WT1 does not obviously change in different inflow conditions. However, the output power of WT2 considering the background turbulence is quite different from that in the uniform inflow.

Under ABL inflow, the power of WT2 is 2.631 MW for 7D and 0° yaw angle, but only 1.098 MW in the uniform inflow. The ambient inflow turbulence intensity makes the power of

WT2 increase by 139.5%, which is obviously due to faster recovery for the wake velocity of WT1. For 7D and 30° yaw angle, the power of WT2 is 4.112 MW, which has exceeded that of WT1. The output power of WT2 is 3.693 MW at uniform inflow, which increases by 11.3%.

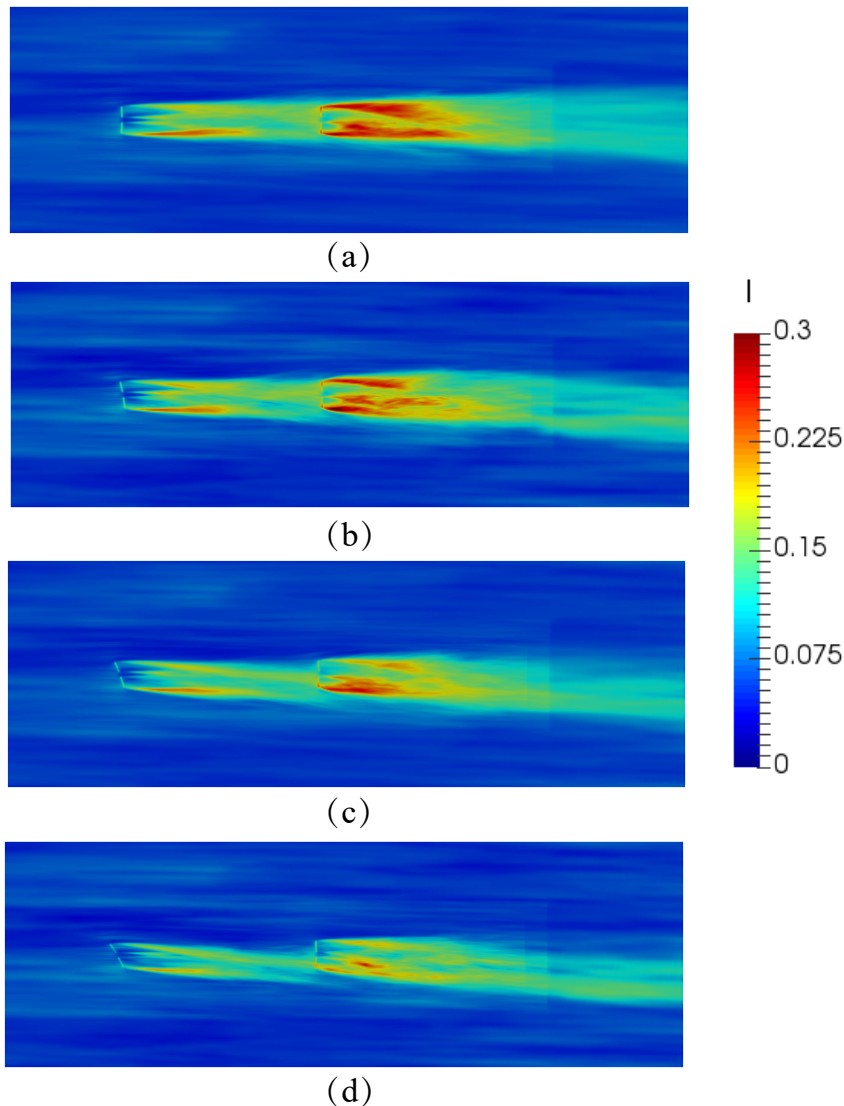

**Figure 18.** Contours of instantaneous turbulence intensity in the horizontal plane at hub height: (**a**) yaw = 0°; (**b**) yaw = 10°; (**c**) yaw = 20°; (**d**) yaw = 30°.

In general, the output power of two turbines in the ABL wind farm is higher than that of the uniform inflow. The background turbulence has little impact on the output power of WT1. However, it will interfere the wake of WT1 to make wake velocity recover faster. Thus, the output power of WT2 is improved due to the increase in the inflow wind speed of WT2. This improvement is particularly obvious in the case of small yaw angles. The optimal yaw angle corresponding to the maximum output power does not vary after considering the background turbulence.

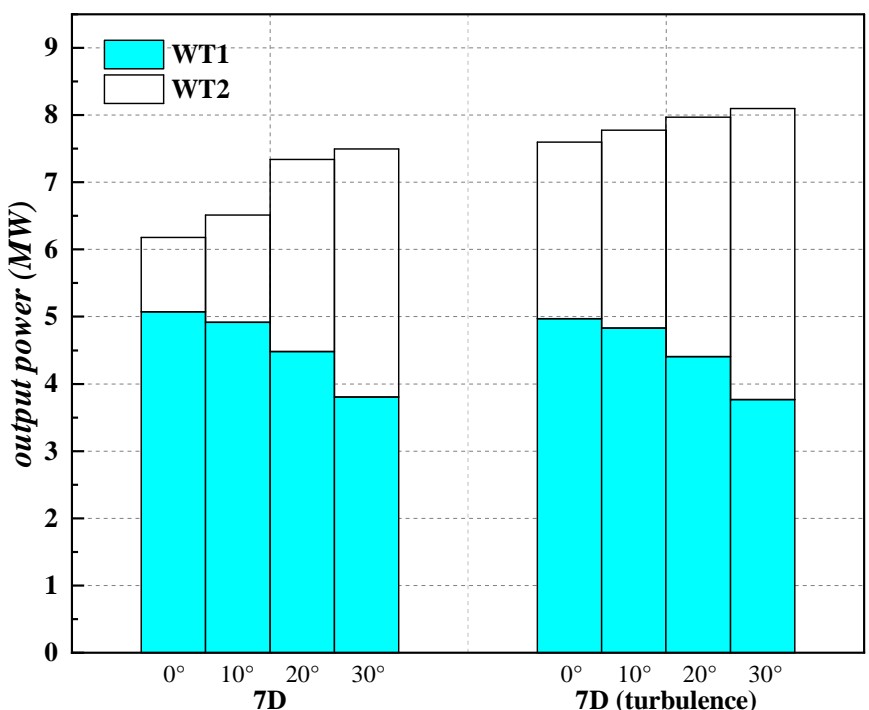

**Figure 19.** The average output power of the turbines in the ABL wind field compared with uniform inflow under 7D conditions.

## 5. Conclusions

Complex wake effects between turbines in a wind farm will significantly affect the output power, blade load, etc. In this study, under the same inflow condition, the wake superposition of two turbines under different yaw and spacing conditions are numerically simulated. The rules of angular speed of the rotors, instantaneous power, and average power with the changes of yaw angle and turbine spacing are analyzed, and the corresponding physical mechanisms are revealed through the wake velocity, turbulence intensity and vortices structures of the wind farm. Finally, the ABL inflow is used, and it is compared with the uniform inflow to explore the effect of yaw control strategy.

(1)  With the yaw-based control of the upstream turbine, the wake of the upstream turbine will be shifted, making the downstream turbine be partially affected by the wake of the upstream turbine, and then the total power of two turbines will increase. When the downstream turbine works in the asymmetric wake, the instantaneous output power fluctuates with a larger amplitude over time. When the spacing is small, the fluctuation of the downstream turbine's output power increases as the yaw angle of the upstream turbine increases. When the spacing becomes large, the increasing trend of fluctuation of the output power occurs in the yaw angle range from 0° to 20°.

(2)  When two turbines have small spacing, a larger yaw angle is needed to obtain greater total power. When the spacing between two turbines is large enough, the increase in yaw angle from 20° to 30° will not significantly increase the total power of the wind farm. The power increase in the downstream turbine due to yaw control cannot offset the power loss of the upstream turbine. In this case, the power output fluctuation of the downstream turbine also increases. When the yaw angle is fixed, the total power increases nonlinearly, and the growth rate decreases with the increase in spacing. In general, the effect of yaw control is greater than that of spacing. Under appropriate spacing and yaw angle conditions, the wind field can obtain the best output power.

(3)  Both velocity deficit and turbulence increase in the wind farm are induced by the upstream turbine. When the upstream turbine has yaw angles, the peak position of its wake velocity profile deviates obviously, which is reset behind the downstream turbine. The larger the spacing between two turbines, the larger the inflow wind speed

can be obtained by the downstream turbine. With the increase in yaw angle, the wake velocity deficit behind the downstream turbine becomes more asymmetrical. In some cases, the maximum wake velocity deficit area moves downstream of the downstream turbine. This is due to the sophisticated interference of wake fields of two turbines.

(4) With the increase in yaw angle, the interaction of vortices becomes more complex, and the asymmetric distribution of vortices intensifies. These vortices interactions make the flow field near the downstream turbine unstable. Thus, the wake velocity deficit and the wind turbine performance are regulated by the asymmetric vortex structures. Therefore, we need to study these situations thoroughly in the actual project.

(5) The background turbulence has little effect on the power of the upstream turbine. The background turbulence increases the inflow velocity of the downstream turbine by reducing the wake effect of the upstream turbine. When the yaw angle of the upstream turbine is small, the effect on the downstream turbine is very significant. The larger the background turbulence intensity, the smaller the yaw angle of the upstream turbine is required to obtain a good comprehensive optimization effect.

In the present study, we studied the total power improvement caused by yaw control under different turbine spacings, quantitatively analyzed the power fluctuation of downstream turbine affected by wake deflection, revealed the interaction mechanism of wake vortices, and compared the results of the uniform inflow with that for the neutral atmospheric boundary layer inflow. In the future, we will consider whether this yaw control strategy is effective under different turbulence intensities.

**Author Contributions:** Writing—review & editing, Z.X. and S.L. (Songyang Liu); Writing—original draft, S.L. (Songyang Liu) and S.L. (Shenghai Liao); Data curation, Z.C.; Software, S.L. (Shenghai Liao) and S.L. (Songyang Liu); Investigation, Z.X. and G.H. All authors have read and agreed to the published version of the manuscript.

**Funding:** This study was financially supported by the National Natural Science Foundation of China (Grant No. 11872174), the Fundamental Research Funds for the Central Universities (Grant No. B200202236) and Key Laboratory of Port, Waterway & Sedimentation Engineering Ministry of Communications, PRC (Yk220001-2).

**Institutional Review Board Statement:** Not applicable.

**Informed Consent Statement:** Not applicable.

**Data Availability Statement:** Not applicable.

**Conflicts of Interest:** The authors declare no conflict of interest.

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
