# Peer review of "Numerical Study on the Yaw Control for Two Wind Turbines under Different Spacings"

_applsci, doi:10.3390/app12147098_

Round 1

Reviewer 2 Report

Dear authors,

Thank you for your work, that provides the literature with an interesting set of large-eddy simulations and analysis of the wind turbine wake dynamics. However, I have a high number of remarks and suggestions, mostly regarding the sentence structures, details about the simulations and higlighting of the result relevance.

Best regards.

l. 18: “The wake effects of wind farm” Do you mean “The wake effects within a wind farm”?

l. 19: How about in site measurements?

l. 20: What do you mean by “the wake model”? Analytical wake models?

l. 21: depends→rely

l. 30: “proposed by some researchers”, be more specific

l. 31: Unclear. Please rephrase.

l. 47: “The development status of the current main simulation methods for wind turbines is briefly introduced in the above sections.” I don’t see the point of this sentence.

l. 51: What is the “above method”?

l. 88-93: This paragraph could be shortened. What do you mean by “the complexity of turbine wake”?

The second and third phrases seem to be redundant.

l. 96: Define “LES” at the first occurrence (l. 37).

l. 97: Why “incompressible”? Isn’t it possible to perform compressible large-eddy simulations? Do you mean that you use an incompressible model? If so, please make a distinction between general information about LES and the specificities of your model. This last remark applies for the whole paragraph.

l.98: “The influences caused by large eddies are directly solved by Navier-Stokes equation,” please rephrase

l. 101: Achieve higher precision and accuracy than what?

l. 109: “by oneEqEddy model” Typo?

l. 118: “The actuator line model simplifies the rotating blades into rotating actuator lines, which several blade elements are distributed along.” Please rephrase.

l. 123: How do you determine lift and drag coefficient? Where do you define the aerodynamic characteristics of the blade?

l. 128: Where does “L” intervene your equations?

l. 136: Avoid “believed”. Replace by “demonstrated that”, “stated that”..

l. 144: What is “the control of Five-Region method”?

l. 152: “this presents” incorrect here.

l. 148-152: Please add details about the simulation and the comparison to the reference. Change the “paper” entry in the legend to something more specific.

l. 165: What is the flow tube effect ? A reference would be welcome.

l. 168: Use “D” to define distances, to be consistent with the rest of the article.

l. 172: “to simulate the tip and root vortices well” → to ensure an accurate simulation of the tip and root vortices?

l. 173: “courant-friedrichs-lewy”, require uppercase for the first letters.

l. 180: Your parameters give a Courant number of 0.114, far below 1 (which you gave as a criteria). Please explain your choice.

Table 2: Your mesh convergence is based on the produced power? Won’t a comparison based on velocity profiles be more relevant?

l. 185: “is not exactly the same”. The reference turbines have a 15cm diameter! It is not the same at all! And there are 5 turbines, not only 2. The optimal yaw angle of 30° is specific to the Bastankhah et al. experiments, not a general value.

l. 271: “Under 5D, 6D, 7D and 8D, the total power increases by 18.31%, 19.20%, 21.29% and 21.99% respectively from 0 to 30 yaw angle.” please rephrase. I suggest: “For the 5D, 6D, 7D and 8D cases, the total power with a 30° yaw angle increases by 18.31%, 19.20%, 21.29% and 21.99% respectively compared to a 0° yaw angle.” Same at l. 276.

l. 283: “For the points in the calculation domain, take the last 200 seconds of the calculation as the time average.” please rephrase.

l. 299: “close to the consistent.” what does this mean?

l. 301: “Generally speaking. The” → “Generally speaking, the”

l. 305: “In addition, the larger the spacing between the two turbines, the wake effect of the upstream turbine on the downstream turbine is relatively weakened, and the faster the wake contour is restored.” please rephrase.

l. 309: “5 to 8 times the rotor diameter” be consistent with the use of “D” on the whole document.

Figure 7: the legend text is too small and the number format nor relevant (2.000e+00)

l. 333: from 1D to 10D downstream WT1

l. 337: reaches 25% - reach 30%

Figure 8: What is the abscissa scale? What is the formula for the “I” calculation? How can we assess the value of “I” from the figure?

l. 342: Be consistent on the whole document, using either “downstream turbine” and “upstream turbine” OR “WT1” and “WT2”.

Figure 9: Same remarks as for Figure 8.

l. 346: Vortex structures.

l. 347: When the spacing between two turbines reaches 7D, the total power is almost the same as at 8D, which is consistent with the work of Yu et al. [29].

Figure 10: How do you explain the wavy shape of the vortices? Is that an interpolation issue related to data visualisation? Is it linked to a low mesh resolution? Have you thought of refining the mesh around the turbines?

l. 370 “the complicated wake effects” please rephrase.

Conclusion: I suggest to make a clear distinction between what you observed that is consistent with the literature, what your study adds to the literature and what are the perspectives for a future work.

1) Further details on the simulation are required (calculation time, number of mesh nodes, computational means, simulated time before the calculation of averages, blade characteristics…).

2) A lot of sentence structures are not idiomatic, which makes the reading difficult. I spotted some of them here-above, but a full revision by a highly-fluent English speaker is required.

3) The authors should emphasize the interest of the study for readers. To which extend the results of this study will help wind turbine operators to control their machine? Is it worth setting up a procedure to increase the yaw of the first turbine in the case when the turbines are perfectly aligned with the wind? (I guess the operators will try to avoid such situations with a proper positioning relative to dominant winds, so this situation should not happen that often?)

4) What could be the influence of ambient turbulence on the results? What would be a typical ambient turbulence intensity? Wouldn’t it reduce the effects studied here, by shortening the WT1’s wake?

Round 2

Reviewer 1 Report

Thank you for taking the time to thoroughly revise your article based on my recommendations. You have adequately addressed all of my major concerns through your revisions, especially the further detail on the validation study and the additional simulations with turbulent conditions.

I have 2 minor suggestions remaining:

1) Although the English language and style is greatly improved, it is still not quite at "native speaker" level. However, it is still very understandable in its current form. Minor English language revisions would be useful, but in my opinion not necessarily required for publication.

2) In the new bar charts for power fluctuations and rotor speed (Figures 6 and 8), I strongly recommend starting the y-axis at 0.0 -- as you do with your output power stacked bar graphs, which are great! The reason for this suggestion is that any bar chart with a non-zeroed y-axis tends to visually exaggerate differences between values. For example, in Figure 8 at 5D distance, the size of fluctuations at 10deg yaw is ~1.7 times the size of the fluctuations at 0deg yaw numerically... but visually the bar is 4-5 times taller. By zeroing the y-axis, the relative size of the bars matches the relative size of the data values.

Reviewer 2 Report

Dear Authors,

I appreciate the additional material provided in the revised version of the article, especially concerning the new validation with ambient turbulence. Regarding the substance of the article, most of my remarks have been addressed (although I still cannot find any information concerning the computational means used for your simulations and the calculation time).

However, as for the first version of the article, there are too many writing issues, from awkward formulations:

l. 40: “For example, Martínez-Tossas et al. [7] proposed the curled wake model, that a curled wake profile is generated in the wind farm, when the wind turbine is yawed.”

l. 42: “too oversimplified”

mistakes:

l. 54: “with the actuator disk model (ALM)” => ALM usually stands for Actuator Line Model

typos:

l. 145: “respectively. we solve”

missing words:

l. 151: “higher precision and accuracy than Reynolds-Averaged Navier Stokes (RANS).” => RANS simulations.

These are just a few examples. I am not going to try spotting them all, as it is not the role of a reviewer (even if I actually played this role in the first review). I will not let this article being published without a significant improvement in the writing, so I suggest that the authors check it again and make it corrected by a highly fluent English speaker.

Best regards.
